# VEC-TOK SPEECH: SPEECH VECTORIZATION AND TOKENIZATION FOR NEURAL SPEECH GENERATION

## ABSTRACT

Language models (LMs) have recently flourished in natural language processing and computer vision, generating high-fidelity texts or images in various tasks. In contrast, the current speech generative models are still struggling regarding speech quality and task generalization. This paper presents Vec-Tok Speech, an extensible framework that resembles multiple speech generation tasks, generating expressive and high-fidelity speech. Specifically, we propose a novel speech codec based on *speech vectors* and *semantic tokens*. Speech vectors contain acoustic details contributing to high-fidelity speech reconstruction, while semantic tokens focus on the linguistic content of speech, facilitating language modeling. Based on the proposed speech codec, Vec-Tok Speech leverages LMs to undertake the core of speech generation. Moreover, Byte-Pair Encoding (BPE) is introduced to reduce the token length and bit rate for lower exposure bias and longer context coverage, improving the performance of LMs. Vec-Tok Speech can be used for intra- and cross-lingual zero-shot voice conversion (VC), zero-shot speaking style transfer text-to-speech (TTS), speech-to-speech translation (S2ST), speech denoising, and speaker de-identification and anonymization. Experiments show that Vec-Tok Speech, built on 50k hours of speech, performs better than other SOTA models.

## 1 INTRODUCTION

In recent years, Large Language Models (LLMs) (Brown et al., 2020; Borsos et al., 2023a) have demonstrated remarkable progress, especially for their excellent zero-shot performance on natural language processing (NLP) tasks. The impressive advances have inspired substantial research extending LLMs to various speech generation tasks, such as text-to-speech (TTS), voice conversion (VC), and speech-to-speech translation (S2ST). With the help of the incredible sequential generation ability of neural language models and unprecedented scale and coverage of audio data, speech generation has been significantly advanced with remarkable zero-shot abilities.

Current large-scale speech generation models usually leverage speech hidden representations to bridge an audio codec and an acoustic model (either an LM (Wang et al., 2023a; Kharitonov et al., 2023) or a diffusion model (Shen et al., 2023)). Specifically, the speech hidden representations can be divided into two categories: *discrete tokens* (Wang et al., 2023a; Kharitonov et al., 2023) and *continuous vectors* (Shen et al., 2023). The discrete tokens are obtained by incorporating vector quantization into the codec for capturing essential information at low bit rates to reconstruct the waveform. However, compressing rich speech information into low-bit-rate representations may degrade the reconstructed speech quality. Therefore, some studies (Zeghidour et al., 2022) propose to utilize Residual Vector Quantizers (RVQ) to learn both linguistic and acoustic features of speech, which contain speech aspects beyond linguistics and may surpass the modeling capacity of conventional LMs. On the other hand, compared to the discrete tokens, the continuous vectors aim to retain all information embedded in speech for high-fidelity speech reconstruction, which brings much pressure to the acoustic model and can not fully utilize the advantages of LMs.

Apart from speech naturalness and quality, *expressiveness* (such as diverse prosody patterns, speaking speed and style, emotions, etc.) is also a critical aspect of human verbal communication. Human speech is a complex signal containing linguistic, non-linguistic, and para-linguistic information (Liu et al., 2020). The linguistic information is the textual content of speech, and para-/non-linguistic characteristics often refer to the speaking style, emphasis, phonation, speaker identity, etc (Nachmani et al., 2023). Previous speech LMs (Wang et al., 2023a; Borsos et al., 2023b) have demonstrated state-of-the-art performance in zero-shot voice cloning by integrating all linguistic-irrelevant acoustic details, including non- and para-linguistic information, from the same speech prompt. However, this paradigm has a limitation for the composition of para-linguistic (e.g., speaking style) and

non-linguistic (e.g., speaker timbre) from different speech prompts - a scenario that is more flexible and practical in real-world speech applications. Specifically, separately adopting speaker prompt and style prompt can enable powerful zero-shot style transfer ability, i.e., a speaker can speak in any designated style given only a few seconds of style reference.

With the aforementioned considerations, we propose **Vec-Tok Speech**, a speech generation model with a novel codec for both continuous **Vec**tor and discrete **Tok**en extraction, with the aim to benefit from both granularities of speech representation. The proposed approach can be applied to multiple downstream tasks, such as text-to-speech, voice conversion, speech-to-speech translation, and Speech denoising. All the above tasks have the powerful adaptation ability in the synthetic speech for arbitrary speakers, and the TTS procedure can simultaneously conduct zero-shot style transfer. The novel codec, named Vec-Tok Codec, consists of an encoder and a decoder, with speech vectors and semantic tokens as intermediate representations. Specifically, *speech vectors* aims to reconstruct the high-fidelity speech, thus containing as many fine speech details as possible. *Semantic tokens* filter out the non-linguistic aspects, i.e., speaker identity, from the speech vector while preserving linguistic and para-linguistic information. An autoregressive LM is employed to produce semantic tokens in downstream tasks. Moreover, we use Byte-Pair Encoding (BPE) (Gage, 1994) to further reduce the token length and the bit rate, thus benefiting the LM from lower exposure bias and longer context coverage. With the semantic token, the target speech can be generated through the codec decoder on the condition of a speech prompt to provide the target speaker identity. The main contributions of this work are summarized as follows:

- *Vec-Tok Codec: A speech codec with high fidelity and low bit rate*. We design a novel speech codec with speech vectorization and tokenization for LM-based speech generation with both high fidelity and low bit rate. Specifically, speech vectorization aims to extract a continuous vector containing as many fine speech details as possible to reconstruct the high-fidelity audio; speech tokenization is to learn the discrete semantic token, preserving linguistic content and style expressions at low bit rates.

- *Vec-Tok Speech: A speech generation framework easily applied to multiple speech generation tasks*. Based on the novel codec, we introduce Vec-Tok Speech, a large speech generation framework that leverages LMs to undertake the core of various downstream tasks, such as TTS and S2ST. Moreover, BPE is introduced to reduce the token length, further benefiting the LM from longer context coverage.

- *Expressive speech generation*. We demonstrate the ability of the proposed Vec-Tok Speech to conduct multi-speaker, multi-style TTS in the zero-shot scenario. A style prompt and a speaker prompt are utilized respectively to provide the style expression for the LM and the speaker identity for the codec decoder, significantly improving the flexibility of a speech-LM in expressive speech generation.

We scale the proposed model to 600 million parameters and 50,000 hours of speech training data. Experiments show that Vec-Tok Speech can generate speech with diverse speaker identities, prosody, and styles in various tasks (VC, TTS, S2ST, etc.) in a zero-shot setup with only a few seconds of speech prompts. Audio samples can be found on our demo page [1].

## 2 RELATED WORK

### 2.1 NEURAL AUDIO CODEC

Neural audio codec (Wu et al., 2023b; Yang et al., 2023a) refers to a kind of neural network model that converts audio waveform into discrete tokens with an encoder and reconstructs audio waveform from these tokens with a decoder. Neural audio codec usually compresses all speech attributes into discrete tokens, such as phase, timbre, prosody, and content. For example, SoundStream (Zeghidour et al., 2022) and Encodec (Défossez et al., 2022) leverage vector-quantized variational auto-encoders (VQ-VAE) with Residual Vector-Quantization (RVQ) technique to compress speech into multiple layers of tokens and have been widely used as intermediate representation in speech and audio generation tasks.

While RVQ-based codecs achieve higher reconstruction quality and lower bitrate than traditional codecs, they are fundamentally tailored for compression and transmission purposes, which makes them suboptimal for intermediate representations for flexible speech generation. Several concerns substantiate this standpoint. First, these codec tokens usually contain as many speech attributes as

---

[1] Demo:https://anonymous.4open.science/w/VecTok/

possible to high-fidelity reconstruction quality, which leads to information redundancy and increases the difficulty of predicting the tokens in downstream tasks. Moreover, the sequence of discrete tokens generated by residual quantizers is often very long, presenting challenges for generative speech models to achieve accurate predictability. This paper proposes a novel speech codec that leverages speech vectors to construct high-quality speech and extracts low-bitrate discrete tokens to ease downstream tasks.

## 2.2 Large scale Generative speech models

A significant shift is observed for generative speech models, where conventional models are predominantly task-specific, trained on distinct datasets tailored to particular objectives. In contrast, the emergence of large-scale generative speech models introduces a paradigm that transcends these limitations, enabling them to tackle a diverse range of novel tasks without explicit training (Le et al., 2023). With the advancement in neural speech codec reviewed in Section 2.1, many recent studies have explored language and diffusion models for speech generation. Encodec stands out for its capacity to extract acoustic tokens among these endeavours. Based on Encodec, VALL-E (Wang et al., 2023a) and its multi-lingual variant VALL-E X (Zhang et al., 2023) achieve zero-shot TTS capabilities by strategically applying autoregressive and non-autoregressive LMs. Meanwhile, SPEAR-TTS (Kharitonov et al., 2023) leverages w2v-BERT (Chung et al., 2021) and SoundStream to derive semantic and acoustic tokens. SPEAR-TTS achieves a multi-speaker TTS system with minimal supervision, integrating multi-stage AR LMs. SoundStorm (Borsos et al., 2023b) extends SPEAR-TTS to more efficient and stable speech generation through masked iterative parallel decoding. On the other hand, models like NaturalSpeech2 (Shen et al., 2023) introduce a text-conditioned diffusion model to generate the latent vectors of the neural audio codec model. TorToise-TTS is a recent speech generation model that combines an autoregressive transformer and a denoising diffusion probabilistic model (DDPM). The autoregressive transformer predicts speech tokens from text, while the DDPM converts speech tokens back to MEL spectrograms. These spectrograms are then sent to a vocoder that synthesizes the final speech. Additionally, a contrastive Language-Voice pretrained (CLVP) module is used to re-score the output of the autoregressive transformer, improving the quality of the synthetic speech.

Beyond TTS, these large-scale generative speech models stretch their capabilities across various tasks. For instance, SpeechGen (Wu et al., 2023a) harnesses a speech LM with diverse prompts to facilitate speech translation, inpainting, and continuation. Similarly, Voicebox (Le et al., 2023) takes a non-autoregressive flow-matching model as its cornerstone, enabling the model to address denoising, zero-shot TTS, and even text-only sampling. This progress of the large-scale speech language model accommodating multiple downstream tasks represents the powerful potential of LM-based variants applying to diverse speech generation tasks. Following this line, we present Vec-Tok Speech, a neural speech generation framework leveraging speech vectorization and tokenization to facilitate various speech generation tasks.

## 3 Method

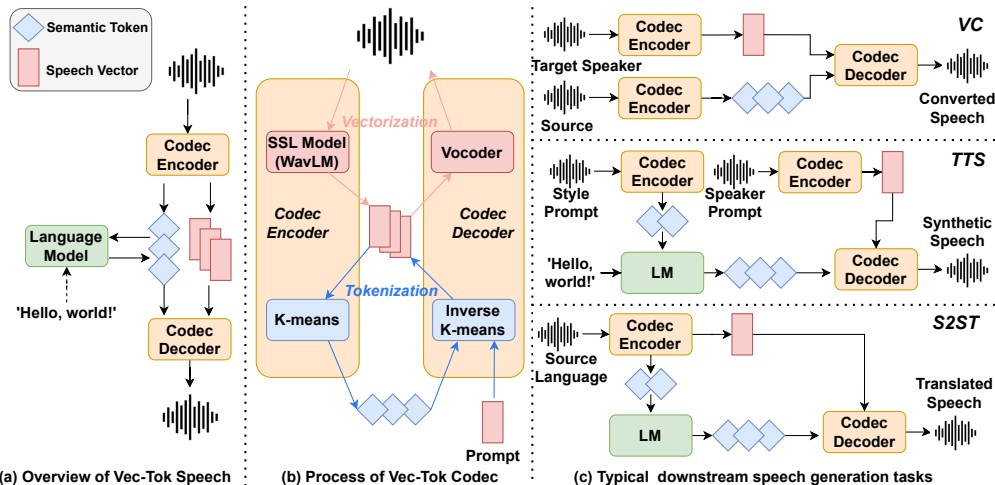

Figure 1: The architecture of Vec-Tok Speech (a) and Vec-Tok Codec (b) and downstream speech generation tasks (c).

## 3.1 OVERVIEW

Figure 1 (a) illustrates the overall structure of our proposed Vec-Tok Speech, which comprises three key components bridged by speech vectors and semantic tokens: a codec encoder, a language model (LM), and a codec decoder. Specifically, the codec encoder is used to extract speech vectors and semantic tokens from speech waveforms, where the semantic tokens are taken as the modeling target of the LM. The codec decoder converts the above semantic tokens and speech vectors back into waveforms. Given the speech waveforms $wave$, the process of the Vec-Tok Speech can be expressed as follows:

$$\{vec, tok\} = \text{Encoder}\left(wave\right), \tag{1}$$

$$\hat{tok} = \text{LM}\left(tok, text\right), \tag{2}$$

$$\hat{wave} = \text{Decoder}\left(vec, \hat{tok}\right), \tag{3}$$

where $vec$ and $tok$ are speech vectors and semantic tokens respectively extracted by the codec Encoder. $\hat{tok}$ is the semantic tokens predicted by the LM; $\hat{wave}$ is the reconstructed waveforms by the codec Decoder.

## 3.2 VEC-TOK CODEC

Figure 1 (b) describes the structure of the Vec-Tok Codec and the speech reconstruction process, where the speech vectors contribute to high-fidelity speech reconstruction and the semantics tokens facilitate language modeling. Specifically, the codec encoder consists of a WavLM model and a K-means module, extracting speech vectors and semantic tokens, respectively. The codec decoder consists of an inverse K-means module and a vocoder, reconstructing speech from semantic tokens and prompt speech vectors.

### 3.2.1 SPEECH VECTORIZATION

To improve the model's high-fidelity speech reconstruction ability, the speech vector is designed to contain as many speech details as possible. Specifically, WavLM (Chen et al., 2022) is adopted to extract speech vectors from speech. It is a self-supervised learning (SSL) model pre-trained on large-scale speech data and performs exceptionally in diverse speech-processing tasks. WavLM extracts fine-grained speech vectors $vec = \{h_i\}_{i=1}^{N}$ from the input wave, where $N$ is the number of frames. Specifically, we take the $6^{th}$ layer of WavLM as the output layer as this layer is proved to contain much acoustic information enabling high-fidelity speech reconstruction (Baas et al., 2023). Then, a neural vocoder is used to reconstruct speech waveforms based on the extracted speech vectors $vec$ since the vocoder can produce waveforms that are nearly indistinguishable from recorded waveforms and are highly generalizable outside of the training set according to TorToise-TTS (Betker, 2023). The process of speech vector extraction and waveform reconstruction is formulated as follows:

$$vec = \text{WavLM}(wave), \tag{4}$$

$$\hat{wave} = \text{Vocoder}(vec), \tag{5}$$

where $vec$ denotes speech vectors extracted by the WavLM, and $\hat{wave}$ is the reconstructed waveforms by the vocoder. The loss functions of the vocoder are as follows:

$$\mathcal{L}_{\text{Vocoder}}^{G} = \sum_{k=1}^{K} \left[\mathcal{L}_{Adv}\left(G; D_k\right) + \lambda_{fm}\mathcal{L}_{FM}\left(G; D_k\right)\right] + \lambda_{vec}\mathcal{L}_{vec}(G)$$

$$\mathcal{L}_{\text{Vocoder}}^{D} = \sum_{k=1}^{K} \mathcal{L}_{Adv}\left(D_k; G\right) \tag{6}$$

where $D_k$ denotes the $k^{th}$ Multi-Period Discriminator (MPD) (Kong et al., 2020) and Multi-Scale Discriminator (MSD) (Kumar et al., 2019) in vocoder, $\mathcal{L}_{FM}$ is the feature matching loss (Larsen et al., 2016) and $\mathcal{L}_{vec}(G)$ is the reconstruction loss of mel-spectrogram.

### 3.2.2 SPEECH TOKENIZATION

Unlike speech vectors, semantic tokens are designed to capture the linguistic content of speech at low bit rates, facilitating downstream language models. First, the speaker-related information is disentangled from the speech vectors to reduce the acoustic information. Then, as the utterance level vector averaged in the time axis strongly correlates with speaker identity (as shown in Appendix A 2), we use utterance level mean normalization on the speech vectors to disentangle speaker-related

information. Finally, we leverage the K-means algorithm to divide speech vectors into $K$ clusters, generating distinct tokens $tok = \{z_i\}_{i=1}^{N}$. As the speech vectors extracted from the WavLM's $6^{th}$ layer are content clustering, tokens obtained from K-means are mainly linguistic-related, thus named semantic tokens. Moreover, we keep the consecutive repeating tokens, which preserves para-linguistic such as rhythm in duration as much as possible. The process of token extraction can be formulated as:

$$tok = \text{K-means}(\text{Normalize}(vec)). \tag{7}$$

Considering that the semantic tokens are extracted from the speech vectors by the K-means algorithm, which inspires us to design an "inverse" K-means model to convert semantic tokens back to speech vectors. Specifically, as the semantic tokens mainly contain linguistic information, we introduce prompt learning during the reconstruction of speech vectors, where a slice of speech vectors containing desired speaker information is used as a prompt to guide the inverse K-means process. To this end, we design a decoder-only model consisting of a look-up table, conformer blocks, and a projection layer, where the K-means table is taken as the look-up table, and the vector of the cluster center is used as the token embedding. The prompt speech vectors are concatenated ahead of the token embeddings as the input of conformer blocks. The self-attention in the conformer blocks is used to capture acoustic information, such as timbre from the prompt speech vectors, as the self-attention module has a global receptive field enabling direct inter-token and token-to-vector interaction. Finally, the output of conformer blocks is sent to the projection layer, generating the final speech vectors. During loss calculation, we mask the prompt part of the predicted speech vector. The above process and the loss function of the inverse K-means model can be defined, respectively as

$$\hat{vec} = \text{Inv\_K}(tok, vec_p), \tag{8}$$

$$\mathcal{L}_{\text{Inv\_K}} = \mathcal{L}_{recon} + \mathcal{L}_{ssim} \tag{9}$$

where $tok$ denotes semantic tokens extracted by the K-means algorithm, $vec_p$ denotes prompt speech vectors containing target speaker information, and $\hat{vec}$ denotes the reconstructed speech vectors. $\mathcal{L}_{recon}$ is the MSE loss between the predicted speech vector and the ground-truth speech vector. Besides, structural similarity $\mathcal{L}_{ssim}$ (Wang et al., 2004) is adopted to measure the similarity between predicted and ground-truth speech vectors. Furthermore, the mask prediction skill is utilized to improve the linguistic comprehension ability of the inverse K-means module. We randomly mask semantic tokens in $\alpha\%$ or replace semantic tokens with another in $\beta\%$, where $\alpha$ and $\beta$ are empirically set to 10.

### 3.3 LANGUAGE MODELS LEVERAGING SEMANTIC TOKENS

Based on the Vec-Tok Codec, an LM is introduced to model the semantic information embedded in speech. Benefiting from the proposed codec, we can model the tokens with only one LM. Specifically, we take two tasks, zero-shot TTS and S2ST, to showcase the effectiveness of our proposed Vec-Tok Codec in a speech LM.

The WavLM extracts the speech vector, and inherently, we get 50 speech tokens per second, which leads to the exposure bias problem and short context length. To alleviate this problem, we use Byte-Pair Encoding (BPE) (Gage, 1994) to compress the length of semantic tokens. Specifically, the BPE algorithm is used to get the vocabulary given the extracted semantic tokens. BPE iteratively merges the most frequent token pairs in semantic tokens to create a vocabulary. With the BPE algorithm, we can effectively compress the length of the token sequence and balance the trade-off between token sequence length and vocabulary size. Thus, during the training and inference stage of LMs, we use BPE-encoded tokens as our input and target.

For both zero-shot TTS and S2ST, we use an autoregressive LM to capture the relationship between the input and output, and the training objective can be formulated as:

$$\mathcal{L}_{\text{TTS}} = -\log \prod_{t=0}^{T} p(z^t | c^{<t}, u; \theta_{\text{TTS}}) \tag{10}$$

$$\mathcal{L}_{\text{S2ST}} = -\log \prod_{t=0}^{T} p(z^t | c^{<t}, z_{src}; \theta_{\text{S2ST}}) \tag{11}$$

where $u$ and $z_{src}$ denote the phoneme sequence and the semantic tokens of source speech, respectively, and $\theta_{\text{TTS}}$ and $\theta_{\text{S2ST}}$ denote the LMs for TTS and S2ST, respectively.

For zero-shot TTS generation, we concat the prompt phonemes, target phonemes, and BPE-encoded prompt semantic tokens as input. Additionally, we follow Tortoise-TTS (Betker, 2023) and train

a CLVP model to re-score the generated tokens. For S2ST, we use the source semantic tokens as input. Conditioned on these, the LM generates the target BPE-encoded semantic tokens. Finally, the codec decoder transforms the semantic tokens into the final waveform.

## 3.4 Applications of Vec-Tok Speech

As shown in Figure 1 (c), we demonstrate the superiority of Vec-Tok Speech by presenting typical examples of creating the context to perform downstream tasks, including VC, TTS, S2ST, Speech denoising, and beyond.

**Zero-shot VC.** Zero-shot VC (Wang et al., 2023b) aims to convert the speaker timbre of any source speech to that of any target speech while maintaining the content of the source speech. We achieve this through the proposed Vec-Tok Codec. Specifically, the speech encoder extracts the semantic token of the source speech and speech vectors of the target speech. Inputting the extracted semantic tokens and speech vectors, the speech decoder generates speech with the speaker timbre of the target speaker and the content of the source speech.

**Zero-shot speaking style transfer TTS.** Zero-shot TTS (Casanova et al., 2022) aims to synthesize the speech with any target speaker' timbre, while zero-shot speaking style transfer TTS refers to the task that synthesizes the speech with not only the timbre of any target speaker but also the speaking style of another speaker. Vec-Tok speech achieves zero-shot speaking style transfer TTS by the following two stages: 1) An LM auto-regressively generates the semantic tokens from the input text and conditions on the prompt semantic tokens. The speaking style of the generated semantic tokens is consistent with the prompt semantic tokens. 2) We input the generated semantic tokens and the speech vectors from another speech prompt representing style to the speech decoder to synthesize the final speech. Thus, Vec-Tok Speech transfers the speaking style through prompt semantic tokens and speaker timbre through prompt speech vectors, synthesizing speech with diverse speaking styles and speaker timbre.

**Speech-to-speech translation.** S2ST (Baldridge, 2004) focuses on translating speech content from a source language to another language. Conventional S2ST is usually based on cascade systems with texts as intermediate representations, including automatic speech recognition, machine translation, and text-to-speech systems, which ignore the transfer of speech prosody and speaker timbre (Song et al., 2023). Ideally, the translated speech should keep the speaker's timbre and even the speaking style in the source language. Vec-Tok speech achieves this target through three stages. In the first stage, we extract the input speech's semantic tokens and speech vectors. In the second stage, a token-to-token LM translates the semantic tokens, which is an implicit translation due to the language-agnostic characteristic of semantic tokens. In the final stage, inputting the translated semantic tokens and speech vectors from the first stage, the speech decoder generates the translated speech. With the prompt speech vectors and the implicit translation, the translated speech maintains both the timbre and speaking style of the source language speaker.

**Speech denoising and bandwidth extension.** Aiming to improve speech perceptual quality, speech denoising refers to noise removal in a speech-noise mixture while bandwidth extension (Andreev et al., 2022) transforms lower sample rate speech, such as 16k Hz speech, to higher sample rate speech, such as 48k Hz speech. In Vec-Tok speech, WavLM performs well in speech denoising, and semantic tokens mainly focus on linguistic contents while ignoring other acoustic aspects such as noise. Therefore, the Vec-Tok Codec can generate clean speech when inputting noise speech. Besides, although the WavLM is pre-trained on a 16k Hz speech corpus, the vocoder can be trained on a 48k Hz speech corpus. Thus, inputting low sample rate speech, Vec-Tok Codec can generate higher sample rate speech with better perceptual quality.

**Speaker de-identification and anonymization.** Speaker de-identification (Yuan et al., 2022) involves removing unique speaker-dependent properties, and speaker anonymization (Srivastava et al., 2022) aims to annoy the speaker to another speaker, which both prevent the speaker from being identified. Vec-Tok Speech can achieve speaker de-identification and anonymization through simple operations on the prompt speech vectors when conducting speech reconstruction. Specifically, we extract speaker-specific speech vectors $V_{spe}$ and corresponding semantic tokens from the speech. Semantic tokens go through the K-means table to generate vectors $V_{agn}$, which is speaker-agnostic. The speech decoder takes $V_{agn}$ and semantic tokens as input to synthesize speaker-agnostic speech, achieving speaker de-identification. Moreover, considering $V_{spe}$ contains speaker-specific characteristics and $V_{agn}$ speaker-agnostic characteristics, we can interpolate between $V_{spe}$ and $V_{agn}$ to obtain a speaker-anonymous speaker vector $V_{ano}$. Thus, the speech decoder takes $V_{ano}$ and semantic tokens as input to synthesize speaker-anonymous speech to accomplish speaker anonymization.

## 4 EXPERIMENTAL SETTINGS

We demonstrate the effectiveness of Vec-Tok Speech through experiments on zero-shot VC, zero-shot speaking style transfer TTS, and S2ST. It is worth noting that Vec-Tok Speech can be easily applied to more tasks, such as Speech denoising, speaker de-identification and anonymization, and demos on these tasks can be found from our online demo page [2].

**Datasets**. The experiments are conducted on 50,000 hours of multi-domain English and Chinese speech recordings. The spoken content is transcribed into International Phonetic Alphabet (IPA) sequences by an Automatic Speech Recognition (ASR) (Yang et al., 2023b) model. The data distribution and preprocessing details are presented in Appendix A.1.

**Model Configuration**. Speech vectors are extracted by a pre-trained WavLM-Large model [3], which produces a single vector for every 20 ms of 16 kHz audio. The number of K-means clusters is set to 300, and the inverse K-means neural module consists of 6 Conformer blocks with 8 attention heads, an embedding dimension of 1024, a feed-forward layer dimension of 4096, and a dropout rate of 0.1. A HiFiGAN V1 model (Kong et al., 2020) is used as the vocoder. The average length of the training waveform is 15 seconds, and the duration of prompt speech is 3 seconds.

The TTS and S2ST models adopt the LLaMA (Touvron et al., 2023) architecture with 12 layers and 12 heads; the hidden and intermediate sizes are 1536 and 6144, respectively. For the BPE encoding, the vocabulary size is 8192, and the token frequency is nearly 16 tokens per second. The CLVP model adopts the BERT architecture with 6 layers, 8 heads, a hidden size of 512, and an intermediate size of 2048 for both text and speech encoder. During inference, the LM generates 256 candidate token sequences and adopts the top 1 sequence as the decoder input based on the CLVP score.

**Model Training**. We use AdamW as the default optimizer and $\beta_1 = 0.9$, $\beta_2 = 0.95$ for all experiments. The LMs for TTS and S2ST are trained by a cosine learning rate schedule on 8 NVIDIA A800 80GB GPUs with a batch size of 64 for each GPU for 10 epochs. The training dataset of the TTS model and the CLVP are the same. The inverse K-means model is trained by 8 NVIDIA 3090 24GB GPUs with a batch size of 12 per GPU for 500k steps. The HiFiGAN model is trained by 4 NVIDIA 3090 24G GPUs with a batch size of 4 per GPU for 2,000k steps.

**Evaluation Metrics**. Both subjective and objective experiments are conducted for a comprehensive evaluation. For subjective evaluation, the mean opinion score (MOS) Zhu et al. (2023) is used to evaluate speech naturalness, speaker similarity, and style similarity. During the scoring process, participants were directed to focus on specific aspects while disregarding others, and thirty participants with basic Chinese-English bilingual skills participated in the subjective experiments.

Speaker cosine similarity (SCS), character error rate (CER), and word error rate (WER) are adopted for objective evaluation. Specifically, SCS is measured by an ECAPA-TDNN model (Desplanques et al., 2020), trained on 3,300 hours of Mandarin speech and 2,700 hours of English speech, totaling 18,083 speakers. We use open-source ASR models provided by the WeNet community Yao et al. (2021) to obtain WER and CER for English and Chinese, respectively. The ASR models are based on the U2++ conformer architecture, trained on 10,000 hours of Gigaspeech English data (Chen et al.) and 10,000 hours of WenetSpeech Mandarin data (Zhang et al., 2022), respectively.

## 5 EXPERIMENTAL RESULTS

### 5.1 ZERO-SHOT VOICE CONVERSION

To evaluate the performance of zero-shot VC, we randomly reserve 50 utterances as source speech and 10 speakers as the target speakers, which are unseen in the training phase. The LM-VC (Wang et al., 2023b) is utilized as the comparison model, which adopts a two-stage framework with three LMs to achieve any-to-any VC. We conduct evaluations on both intra- and cross-lingual zero-shot VC.

Table 1 shows the results of intra-lingual zero-shot VC. Compared to LM-VC, Vec-Tok Speech achieves superior speech naturalness and better intelligibility with lower WER/CER. This indicates that Vec-Tok Speech effectively preserves the linguistic content of source speech and produces nat-

---

[2]Demo:https://anonymous.4open.science/w/VecTok/
[3]WavLM:https://huggingface.co/microsoft/wavlm-large

ural speech. The speaker similarity MOS and SCS results verify the excellent ability to capture the target speaker's timbre in a zero-shot manner. Moreover, we validate the effectiveness of the inverse K-means model by replacing semantic tokens with frame-level phoneme sequences. Specifically, a TDNN-F (Peddinti et al., 2015) model trained on WenetSpeech (Zhang et al., 2022) and GigaSpeech (Chen et al.) is adopted to provide the frame-level phoneme sequence from speech. From the result, we can observe a sharp decline in speaker similarity MOS, demonstrating the effectiveness of the inverse K-means idea. In addition, we replace WavLM with XSLR [4] (Babu et al., 2022), a wav2vec 2.0 model pre-trained on a multilingual dataset, to investigate the influence of self-supervised learning models on performance. We observe the performance of XLSR is similar to that of WavLM. Based on SUPERB (Yang et al., 2021), we conjecture that the shallow-layer feature can generalize well to various speech inputs.

Table 1: Results of inter-lingual zero-shot VC with 95% confidence interval.

| Model | Naturalness MOS($\uparrow$) | Speaker Similartiy MOS($\uparrow$) | CER%($\downarrow$) | WER%($\downarrow$) | SCS($\uparrow$) |
|---|---|---|---|---|---|
| GroundTruth | $4.42 \pm 0.07$ | - | 2.8 | 1.9 | - |
| LM-VC | $3.78 \pm 0.10$ | $3.74 \pm 0.08$ | 3.2 | 2.7 | 0.892 |
| Vec-Tok Speech | $3.86 \pm 0.10$ | $\mathbf{3.93 \pm 0.09}$ | 3.0 | 2.6 | 0.927 |
| w/o Inv_k | $\mathbf{3.88 \pm 0.08}$ | $3.67 \pm 0.08$ | **2.9** | **2.4** | 0.878 |
| w XLSR | $3.85 \pm 0.11$ | $3.89 \pm 0.07$ | 3.0 | 2.7 | **0.928** |

Table 2 shows the results of cross-lingual zero-shot VC. Notably, "GroundTruth" represents the source speech recording. We can see that there is an apparent decline in all evaluation metrics compared to those in the intra-lingual zero-shot VC in Table 1, demonstrating the challenge of cross-lingual zero-shot VC. However, Vec-Tok Speech obviously outperforms LM-VC, showing its extraordinary ability to achieve native pronunciation while maintaining the speaker timbre for foreign speakers in zero-shot VC.

Table 2: Results of cross-lingual zero-shot VC with 95% confidence interval.

| Model | Naturalness MOS($\uparrow$) | Speaker Similartiy MOS($\uparrow$) | CER%($\downarrow$) | WER%($\downarrow$) | SCS($\uparrow$) |
|---|---|---|---|---|---|
| GroundTruth | $4.42 \pm 0.07$ | - | 2.8 | 1.9 | - |
| LM-VC | $3.60 \pm 0.11$ | $3.66 \pm 0.08$ | 3.9 | 3.1 | 0.886 |
| Vec-Tok Speech | $\mathbf{3.81 \pm 0.10}$ | $\mathbf{3.90 \pm 0.09}$ | **3.2** | 2.9 | 0.919 |
| w/o Inv_K | $3.79 \pm 0.09$ | $3.63 \pm 0.10$ | 3.4 | **2.8** | 0.871 |
| w XLSR | $3.80 \pm 0.08$ | $3.87 \pm 0.09$ | 3.1 | 3.0 | **0.922** |

## 5.2 Zero-shot text-to-speech

The proposed method can generate English and Chinese speech on the condition of an unseen speaker prompt and an unseen style prompt, enabling stylistic speech generation for a speaker. But to the best of our knowledge, the current LM-based zero-shot TTS models can only condition on a single speech prompt representing both speaker and speaking style. Hence, we first experiment on the zero-shot TTS task utilizing the speaker prompt as the style prompt simultaneously for a fair comparison. Specifcially, the SOTA multilingual zero-shot TTS models, VALL-E X (Zhang et al., 2023) and Bark [5] are involved in this evaluation. VALL-E X is trained on the same corpus as our Vec-Tok Speech and the Bark model's official checkpoint is directly used in the experiment.

Table 3 shows the evaluation results of zero-shot TTS. As we can see, there is a trade-off between the naturalness and speaker similarity in VALL-E X and Bark. However, Vec-Tok Speech outperforms them in terms of both naturalness and speaker similarity, demonstrating the effectiveness of speech vectorization for natural speech generation and tokenization for linguistic information capturing. Besides, Vec-Tok Speech obtains better intelligibility as well with lower CER/WER in Chinese/English. Notably, VALL-E X and Bark obtain 5.3% and 5.8% CERs, respectively, on the Chinese testing clips, which are much higher than those obtained by Vec-Tok Speech. Moreover, Vec-Tok speech achieves the highest SCS score, consistent with speaker similarity MOS. These results indicate the superior zero-shot TTS performance of the proposed method for unseen target speakers.

---

[4]XSLR:https://huggingface.co/facebook/wav2vec2-xls-r-300m
[5]Bark:https://github.com/suno-ai/bark

Further assessment of Vec-Tok Speech is conducted with the ablation of the BPE and CLVP. As shown in Table 3, removing the BPE results in a sharp decline in naturalness MOS, WER, and CER. Notably, the listeners argue that major problems for the model without BPE lie in the synthetic long utterances. The results indicate that lower exposure bias and longer context benefit speech naturalness. This result shows lower exposure bias and longer context length benefit natural pronunciation. In addition, when removing the CLVP, we found that the extremely poor samples would affect the evaluation results to a large extent. Therefore, there is a decline in performance when the number of candidate token sequences from the CLVP is reduced. To further validate the ability of zero-shot style transfer, two different speech prompts, separately providing speaker timbre and style expressions, are conditioned to Vec-Tok Speech for TTS. Specifically, we add 20 utterances of speech with obvious speaking style as prompts in LM and evaluate the style similarity of the synthetic speech. Both subjective and objective results indicate that Vec-Tok Speech has the zero-shot ability to simultaneously transfer the speaking style and speaker timbre from different speech prompts.

Table 3: Results of zero-shot TTS with 95% confidence interval.

| Model | Naturalness MOS(↑) | Speaker Similarity MOS(↑) | Style Similarity MOS(↑) | CER%(↓) | WER%(↓) | SCS(↑) |
|---|---|---|---|---|---|---|
| Ground Truth | 4.40 ± 0.07 | - | - | 2.8 | 1.9 | - |
| VALL-E X | 3.72 ± 0.12 | 3.67 ± 0.10 | - | 5.3 | 3.9 | 0.852 |
| Bark | 3.68 ± 0.11 | 3.73 ± 0.12 | - | 5.8 | 3.6 | 0.866 |
| Vec-Tok Speech | **3.92 ± 0.08** | 3.87 ± 0.07 | - | **3.7** | **3.4** | **0.909** |
| w/ Style Prompt | 3.88 ± 0.10 | 3.84 ± 0.08 | **3.87 ± 0.10** | 3.9 | 3.7 | 0.902 |
| w/o BPE | 3.80 ± 0.10 | **3.88 ± 0.08** | - | 4.6 | 4.2 | 0.908 |
| w/ CLVP-64 | 3.84 ± 0.12 | 3.86 ± 0.07 | - | 4.4 | 4.1 | 0.907 |

### 5.3 SPEECH-TO-SPEECH TRANSLATION

We assess the performance of our model in the S2ST task on the test set of GigaST (Ye et al., 2022) in terms of naturalness, speaker similarity, BLEU, and SCS. We utilize the constrained En-Zh speech translation model, involving the same training data and similar parameter size in GigaST (Ye et al., 2022) as the compared method.

As shown in Table 4, Vec-Tok Speech reaches a comparative BLEU score compared to the cascaded translation model in GigaST that translates English speech into Chinese text. Speaker similarity MOS and cosine similarity scores both indicate the generated speech preserves the speaker timbre of the source speech during translation. Vec-Tok Speech achieves a naturalness MOS of 3.69, demonstrating that the proposed method can effectively synthesize natural speech of the target language. These results imply that the proposed method can conduct reasonable speech-to-speech translation while preserving the speaker timbre of the source language.

Table 4: Results of S2ST with 95% confidence interval.

| Model | Naturalness MOS(↑) | Speaker Similarity MOS(↑) | BLEU(↑) | SCS(↑) |
|---|---|---|---|---|
| GroundTruth | 3.87 ± 0.08 | - | - | - |
| GigaST | - | - | **22.30** | - |
| Vec-Tok Speech | **3.69 ± 0.12** | **3.85 ± 0.09** | 21.56 | **0.904** |

## 6 CONCLUSION

This paper presents Vec-Tok Speech, a framework that resembles multiple speech generation tasks and generates expressive and high-fidelity speech. Specifically, we propose a novel speech codec based on speech vectors and semantic tokens, named Vec-Tok Codec. Specifically, speech vectors contain acoustic details contributing to high-fidelity speech reconstruction, while semantic tokens focus on the linguistic content of speech, serving effective language modeling. Based on the proposed speech codec, Vec-Tok Speech leverages LMs to undertake the core of speech generation. Moreover, Byte-Pair Encoding is further introduced to reduce the token length and the bit rate, thus benefiting the LM from lower exposure bias and longer context coverage. Vec-Tok speech can be used for various tasks, including intra- and cross-lingual zero-shot voice conversion, zero-shot speaking style transfer text-to-speech, speech-to-speech translation, speech denoising and beyond. Extensive experiments demonstrate the good design of Vec-Tok Codec and Vec-Tok Speech.

## 7 REPRODUCIBILITY STATEMENT

To help readers reproduce our experiments, we provided descriptions of our architectures in Section 3, and details of our models about hyperparameters in Section 4. We also provide details of our data processing pipeline in Appendix A.1. Moreover, we plan to release our source code to ensure the reproducibility of this paper.

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

# A APPENDIX

## A.1 DATA DISTRIBUTION AND PROCESSING

Table 5 shows detailed data distribution, where WenetSpeech (Zhang et al., 2022), GigaSpeech (Chen et al.), LibriTTS (Zen et al., 2019), GigaST (Ye et al., 2022) and GigaS2S (Bytedance, 2023) are all open-source datasets downloaded from the Internet. Furthermore, a 20k-hour expressive audiobook dataset XBook is collected from internal resources, and a clean subset named XBook-clean is selected. Specifically, we select 1,500 hours of speech from XBook-clean and GigaSpeech to train the K-means model and the vocoder, while the inverse K-means model is trained on the whole XBook dataset.

Data processing consists of three main modules: speech recognition, speech denoising, and text-to-IPA. Specifically, the voice activity detection (VAD) result, timestamp, and text transcription are obtained from waveforms by an ASR model (Yang et al., 2023b), where the text transcription is processed to obtain the corresponding IPA-level transcription by a text front-end, including text normalization, word segmentation, prosody prediction, polyphone disambiguation, g2p, etc. On the other hand, denoised waveforms are acquired by the DeepfilterNet2 (Schröter et al., 2022) network. All denoised waveforms are downsampled to 16k Hz.

Table 5: Datasets used to train Vec-Tok Speech.

| Datasets | Language | Duration (hours) | Usage |
|---|---|---|---|
| WeNetSpeech (Zhang et al., 2022) | Chinese | 10k | VC |
| GigaSpeech (Chen et al.) | English | 9k | VC & S2ST & TTS |
| GigaS2S (Ye et al., 2022; Bytedance, 2023) | Chinese | 9k | S2ST |
| LibriTTS (Zen et al., 2019) | English | 580 | TTS |
| XBook | Chinese | 20k | VC |
| XBook-clean | Chinese | 8k | TTS |

## A.2    CORRELATION BETWEEN SPEAKER ID AND WAVLM FEATURE

We randomly select 30 speakers from VCTK (Veaux et al., 2016), 100 utterances per speaker, to verify the relationship between the speaker identity and the WavLM features. First, the VAD detects the vocal segments sent to WavLM to extract the speech vectors. Then, we calculate the averaged vectors in the time axis for each utterance and visualize them through t-SNE. As shown in Figure 2, averaged vectors are well clustered by speaker identity, demonstrating the strong correlation between the averaged vectors and the speaker identities.

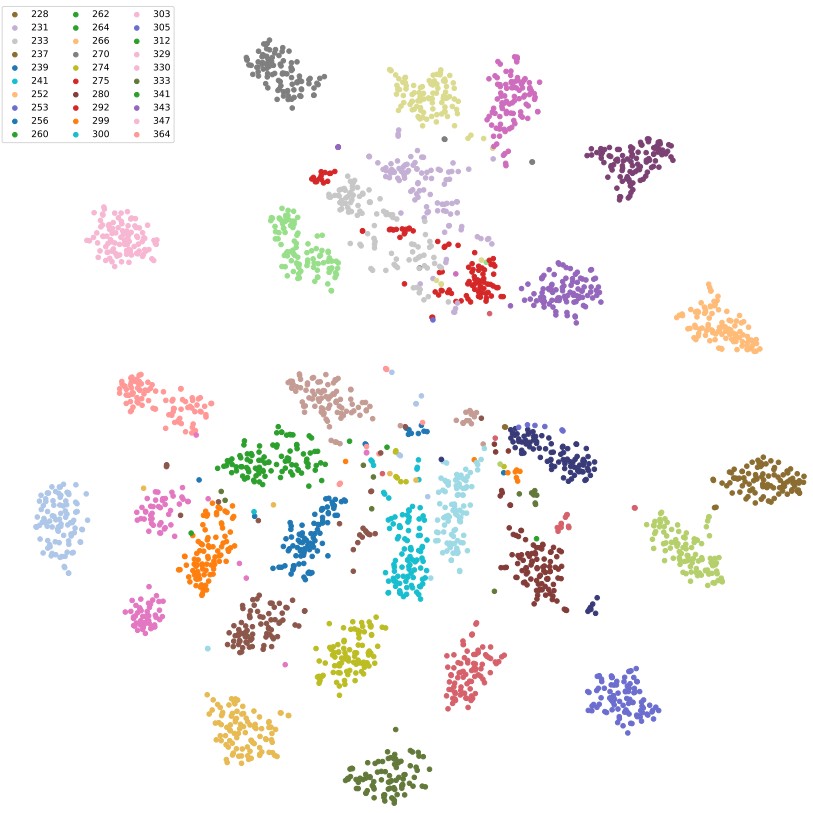

Figure 2: T-SNE visualization of the averaged WavLM features for 30 speakers in the VCTK dataset.

## A.3 QUANTITATIVE STUDY OF SPEECH DENOISING

We evaluate the proposed framework in speech denoising task. We use the synthetic test set with reverberation of DNS challenge-Interspeech 2021 as our test set, and we follow the evaluation metric of DNS challenge 2021 Reddy et al. (2021). Additionally, we compare our result with some models trained explicitly with denoising objectives. The results are shown in Table.6.

Table 6: The DNSMOS result of tested denoising systems

| Model | DNSMOS | | |
|---|---|---|---|
| | SIG | BAK | OVL |
| Noisy [2] | 1.760 | 1.497 | 1.392 |
| Conv-TasNet [3] | 2.415 | 2.710 | 2.010 |
| Demucs [4] | 2.510 | 2.641 | 2.215 |
| Inter-SubNet [5] | 2.651 | 2.581 | 2.362 |
| CDiffuSE [6] | 2.541 | 2.300 | 2.190 |
| Vec-Tok Speech | 2.681 | 2.912 | 2.235 |

## A.4 QUANTITATIVE STUDY OF CODEC RECONSTRUCTION QUALITY

We test the PESQ and STOI metrics in 100 speech clips from our test set, and compare the reconstruction performance with Encodec [6] Défossez et al. (2022). The results are shown in Table.7.

Table 7: Speech reconstruction performance of Vec-Tok Speech codec

| Model | bitrate | PESQ | STOI |
|---|---|---|---|
| Encodec | 1.5kbps | 1.42 | 0.84 |
| Encodec | 3kbps | 1.87 | 0.90 |
| WavLM feature +Vocoder | - | 2.50 | 0.95 |
| WavLM feature +KMeans300 +Inverse k-means +Vocoder | $\sim$0.41kbps | 1.93 | 0.92 |

The results indicate that our codec is capable of reconstructing speech with a low bitrate and minor loss of sound quality.

---

[6]https://github.com/facebookresearch/encodec

