# OpenReview forum: "Vec-Tok Speech: Speech Vectorization and Tokenization for Neural Speech Generation"
_ICLR.cc/2024/Conference — Submitted to ICLR 2024_

### Official Review · Reviewer_zJWA · 2023-10-28

**Soundness:** 2 fair
**Presentation:** 3 good
**Contribution:** 3 good
**Rating:** 3
**Confidence:** 4

**Summary:**

This work introduces VEC-TOK SPEECH, an extensive framework for a variety of speech generation tasks. The framework integrates an innovative codec that is capable of disentangling linguistic content and acoustic details from speech, as well as a language model tailored for conditional semantic token generation. Moreover, the authors ease the token sequence length issue by implementing the BPE technique. The experimental results confirm the effectiveness of the proposed approach.

**Strengths:**

1.	This work introduces a novel paradigm: representing relatively simple linguistic content with a single semantic token sequence, while capturing complex audio details using continuous vectors, as opposed to the multiple acoustic token sequences in previous works. This design can lessen the challenges of language modeling and enhance generation quality.
2.	The article consolidates multiple speech generation tasks within a single, concise, and scalable framework.

**Weaknesses:**

1.	The paper may need more experimental results to improve its soundness, including:
a) The state-of-the-art (SOTA) model for cross-lingual zero-shot TTS is not VALL-E X, but Mega-TTS[1]. Including such a comparison would make the paper sounder.
b) The reconstruction quality of the proposed codec is not assessed thoroughly.
2.	The quality of the provided samples is not satisfactory when compared with the samples from [1] (demo link: https://mega-tts.github.io/demo-page/).
3.	The abstract contains some overstatements. Indeed, this work is not the first work to unify multiple speech generation task ([1-4]).
        `` "In contrast, the current speech generative models are still struggling regarding speech quality and task generalization."``

4.	The use of BPE for semantic tokens was first proposed in [5]. Including the reference and a discussion on this topic may be necessary.

[1] Jiang, Z., Ren, Y., Ye, Z., Liu, J., Zhang, C., Yang, Q., Ji, S., Huang, R., Wang, C., Yin, X., Ma, Z., & Zhao, Z. (2023). Mega-TTS: Zero-Shot Text-to-Speech at Scale with Intrinsic Inductive Bias. ArXiv, abs/2306.03509.

[2] Shen, K., Ju, Z., Tan, X., Liu, Y., Leng, Y., He, L., Qin, T., Zhao, S., & Bian, J. (2023). NaturalSpeech 2: Latent Diffusion Models are Natural and Zero-Shot Speech and Singing Synthesizers. ArXiv, abs/2304.09116.

[3]  Huang, R., Zhang, C., Wang, Y., Yang, D., Liu, L., Ye, Z., Jiang, Z., Weng, C., Zhao, Z., & Yu, D. (2023). Make-A-Voice: Unified Voice Synthesis With Discrete Representation. ArXiv, abs/2305.19269.

[4] Le, M., Vyas, A., Shi, B., Karrer, B., Sari, L., Moritz, R., Williamson, M., Manohar, V., Adi, Y., Mahadeokar, J., & Hsu, W. (2023). Voicebox: Text-Guided Multilingual Universal Speech Generation at Scale. ArXiv, abs/2306.15687.

[5] Chou, J., Chien, C., Hsu, W., Livescu, K., Babu, A., Conneau, A., Baevski, A., & Auli, M. (2023). Toward Joint Language Modeling for Speech Units and Text. ArXiv, abs/2310.08715.

**Questions:**

1. Does the SCS metric measure the similarity between the generated audio and the original recording or the reconstructed samples?
2. Do consecutive repeating tokens impact the BPE efficiency? Could you provide the average token sequence length after deduplicating the tokens?

**Details Of Ethics Concerns:**

None.

---

> ### Author Response · Authors · 2023-11-22
> **Author Response to Reviewer zJWA (1/2)**
>
> Q: Does the SCS metric measure the similarity between the generated audio and the original recording or the reconstructed samples?
>
> A: The SCS metric measures the similarity between the generated audio and the original recording.
>
> Q: Do consecutive repeating tokens impact the BPE efficiency? Could you provide the average token sequence length after deduplicating the tokens?
>
> A: Consecutive repeated tokens can affect the efficiency of the BPE algorithm. Nevertheless, these repeated tokens carry speech prosody such as rhythm [1].  Moreover, when we remove the repeated tokens, the language model would not be able to model the rhythms of speech, and we must use a duration module to model the rhythm. This poses a challenge to the Non-Autoregressive inverse K-means module. As a result, we have not provided the average length of the token sequence after deduplication.
>
> Q: The reconstruction quality of the proposed codec is not assessed thoroughly.
>
> A: We are sorry for insufficient training in vocoder, which leads to low reconstruction quality. We have supplemented the speech reconstruction demo and qualitative results of reconstruction quality, which proves our codec is effective. We test the PESQ and STOI metrics in 100 speech clips from our test set. The results are shown at next comment.
>
> The results indicate that our codec is capable of reconstructing speech with a low bitrate and minor loss of sound quality.
>
> Q: The paper may need more experimental results to improve its soundness, including: a) The state-of-the-art (SOTA) model for cross-lingual zero-shot TTS is not VALL-E X, but Mega-TTS[1]. Including such a comparison would make the paper sounder.
>
> Q: The quality of the provided samples is not satisfactory when compared with the samples from [1] (demo link: https://mega-tts.github.io/demo-page/).
>
> A: Mega-TTS [2] is a powerful zero-shot TTS model. However, the MRTE and PLM modules in Mega-TTS require speaker ID during the training phase, which is not available in our large-scale crawled dataset, and labelling the entire dataset with a speaker diarization model and extracting their phoneme-level alignments surpasses our limited resources' capacity.
>
> Moreover, according to the [review guideline](https://iclr.cc/Conferences/2024/ReviewerGuide) of ICLR,
> > if a paper was published (i.e., at a peer-reviewed venue) on or after May 28, 2023, authors are not required to compare their own work to that paper.
>
> And due to limited resources, we have not invested our resources in the replication of Mega-TTS.
>
> Q: The abstract contains some overstatements. Indeed, this work is not the first work to unify multiple speech generation task. "In contrast, the current speech generative models are still struggling regarding speech quality and task generalization."
>
> A: We clarify that a unified framework overriding more speech generation tasks is attracting attention [4,5]. However, these speech-generative models obtain good speech quality while the number of speech-generation tasks is limited.
>
> Q: The use of BPE for semantic tokens was first proposed in [3]. Including the reference and a discussion on this topic may be necessary.
>
> A: The article [3] became available on Arxiv on 12th October 2023. Unfortunately, we are unable to provide the reference or a discussion due to the ICLR submission deadline of 28th September 2023. So we will discuss this article here.
>
> In the article [3], they use the deduplicated final layer representation of HuBERT base model,  in which the operation of deduplication leads to the loss of rhythmic information[1], and the final layer representation leads to worse pitch and energy reconstruction accoding to [6]. Therefore, this token will be difficult to reconstruct, and thus this representation is not suitable for speech generation tasks.
>
>
>
> [1] Song, K., Ren, Y., Lei, Y., Wang, C., Wei, K., Xie, L., Yin, X., & Ma, Z. (2023). StyleS2ST: Zero-shot Style Transfer for Direct Speech-to-speech Translation.ArXiv. /abs/2305.17732
>
> [2] Jiang, Z., Ren, Y., Ye, Z., Liu, J., Zhang, C., Yang, Q., Ji, S., Huang, R., Wang, C., Yin, X., Ma, Z., & Zhao, Z. (2023). Mega-TTS: Zero-Shot Text-to-Speech at Scale with Intrinsic Inductive Bias. ArXiv, abs/2306.03509.
>
> [3] Chou, J., Chien, C., Hsu, W., Livescu, K., Babu, A., Conneau, A., Baevski, A., & Auli, M. (2023). Toward Joint Language Modeling for Speech Units and Text. ArXiv, abs/2310.08715.
>
> [4] Le, M., Vyas, A., Shi, B., Karrer, B., Sari, L., Moritz, R., Williamson, M., Manohar, V., Adi, Y., Mahadeokar, J., & Hsu, W. (2023). Voicebox: Text-Guided Multilingual Universal Speech Generation at Scale. ArXiv, abs/2306.15687.
>
> [5] Wu, H., Chang, K., Wu, Y., & Lee, H. (2023). SpeechGen: Unlocking the Generative Power of Speech Language Models with Prompts.ArXiv. /abs/2306.02207
>
> [6] G.-T. Lin, C.-L. Feng, W.-P. Huang, Y. Tseng, T.-H. Lin, C.-A. Li, H.-y. Lee, and N. G. Ward, “On the utility of self-supervised models for prosody-related tasks,” in IEEE SLT, 2023.

---

> ### Author Response · Authors · 2023-11-22
> **Author Response to Reviewer zJWA (2/2)**
>
> Reconstruction quality of our codec:
> | Model | bitrate | PESQ | STOI |
> | --- | --- | --- | --- |
> | Encodec | 1.5kbps | 1.42 | 0.84 |
> | Encodec | 3kbps | 1.87 | 0.90 |
> | WavLM feature +Vocoder | - | 2.50 | 0.95 |
> | WavLM feature +KMeans300 +Inverse k-means +Vocoder | ~0.41kbps | 1.93 | 0.92 |

---

> ### Comment · Reviewer_zJWA · 2023-11-22
>
> Thank you for your response. While some issues have been addressed, I still think the audio quality is not good enough, and recommend additional revisions to make the paper more sound. Therefore, I would like to keep my initial score unchanged.

---

### Official Review · Reviewer_KgQK · 2023-10-31

**Soundness:** 3 good
**Presentation:** 3 good
**Contribution:** 3 good
**Rating:** 5
**Confidence:** 4

**Summary:**

This paper presents a novel speech generation model called Vec-Tok Speech with a codec  leveraging speech vectorization and tokenization to facilitate various speech generation tasks. The proposed architecture is beneficial for both high-fidelity speech reconstruction and accurate linguistic content of speech.
The paper further introduces Byte-Pair Encoding technique in order to reduce the token length and bit rate for lower exposure bias and longer context coverage, thus improving the performance of language models.

**Strengths:**

1. This paper tackles the shortcomings of the previous works related to neural audio codec, RVQ-based codecs, which leads to information redundancy and increases the difficulty of predicting the tokens in downstream tasks, by proposing Byte Pair Encoding to compress the length of semantic tokens.
2. The authors have conducted extensive amounts of downstream tasks, showing the superiority of Vec-Tok Speech: Zero-shot VC, Zero-shot speaking style transfer TTS, Speech to Speech translation, Speech denoising and bandwidth extension, and speaker de-identification and anonymization.
3. This paper is well structured and easy to read.

**Weaknesses:**

1. The proposed model seems to be the combination of the existing technique: vectorization, tokenization, and BPE, without any novel architecture.
2. I am not sure it is novel enough to adapt Byte-Pair Encoding technique to compress the length of semantic tokens. Many existing works have leveraged the BPE technique in audio processing [1,2], so it is better to distinguish the proposed model from the ones that have utilized the BPE.
3. I think even though the authors have presented various amount of downstream task, the performances of previous works that they are comparing with seem to be limited overall. For example in zero-shot tts, why not comparing with Voicebox?
4. Moreover, there is no quantitative performance on speech denoising. Although the results are shown in the demo page, it is better to report the qualitative results with metrics like MCD, STOI, PESQ for performance reporting.

[1] Elkahky, Ali, et al. "Do Coarser Units Benefit Cluster Prediction-Based Speech Pre-Training?." ICASSP 2023-2023 IEEE International Conference on Acoustics, Speech and Signal Processing (ICASSP). IEEE, 2023.

[2] Algayres, Robin, et al. "Generative Spoken Language Model based on continuous word-sized audio tokens." arXiv preprint arXiv:2310.05224 (2023).

**Questions:**

Please refer to the weakness.

---

> ### Author Response · Authors · 2023-11-22
> **Author Response to Reviewer KgQK**
>
> Q: The proposed model seems to be the combination of the existing technique: vectorization, tokenization, and BPE, without any novel architecture.
>
> A: The techniques already exist. We explore and introduce the pre-trained self-supervised model and the idea of inverse k-means into speech codec. We decouple speech into linguistic, para-linguistic, and non-linguistic features, trying to simplify the speech tokens for better performance of speech LMs. Thus, we can stimulate the potential of speech LM and build a speech generation framework easily applied to multiple speech generation tasks.
>
> Q: It is better to distinguish the proposed model from the ones that have utilized the BPE.
>
> A: Thank you for your suggestion. BPE has been investigated in the pre-training speech model [1] where BPE, deduplication, DP smoothing and various techniques for a less redundant representation. We explore the utilization of BPE in speech-generation tasks. Specifically, directly applying BPE to acoustic tokens leads to different sequence lengths for different layers of acoustic tokens. Applying BPE to semantic tokens benefits the speech LM from lower exposure bias and longer context coverage, which reduces the accumulative error due to the nature of the autoregressive language model.
>
> Q: The performances of previous works that they are comparing with seem to be limited overall. For example in zero-shot tts, why not comparing with Voicebox?
>
> A: The large-scale speech generation model is developing rapidly, but there is very little open-source work or benchmark. Besides, the dataset used in each work is very different. This brings us a great challenge to reproduce. We successfully reproduced VALLE-X and other models with open-source implementations but failed in Voicebox.
>
> Q: Moreover, there is no quantitative performance on speech denoising. Although the results are shown in the demo page, it is better to report the qualitative results with metrics like MCD, STOI, PESQ for performance reporting.
>
> A: Thank you for your suggestion. We have evaluated our framework in the speech denoising task. We use the synthetic test set with reverberation of DNS challenge-Interspeech 2021 as our test set, and we follow the evaluation metric of DNS challenge 2021 [2]. Additionally, we compare our result with some models trained explicitly with denoising objectives [3-6]. The results are shown as follows.
>
> | Model | DNSMOS| | |
> | --- | --- | --- | --- |
> |  | SIG | BAK | OVL |
> | Noisy [2] | 1.760 | 1.497 | 1.392 |
> | Conv-TasNet [3] | 2.415 | 2.710 | 2.010 |
> | Demucs [4] | 2.510 | 2.641 | 2.215 |
> | Inter-SubNet [5] | 2.651 | 2.581 | 2.362 |
> | CDiffuSE [6] | 2.541 | 2.300 | 2.190 |
> | Vec-Tok Speech | 2.681 | 2.912 | 2.235 |
>
>
> [1] Elkahky, Ali, et al. "Do Coarser Units Benefit Cluster Prediction-Based Speech Pre-Training?." ICASSP 2023-2023 IEEE International Conference on Acoustics, Speech and Signal Processing (ICASSP). IEEE, 2023.
>
> [2] Chandan K.A. Reddy et al., “INTERSPEECH 2021 Deep Noise Suppression Challenge,” in Proc. Interspeech, 2021, pp. 2796–2800.
>
> [3] Yi Luo and Nima Mesgarani, “Conv-tasnet: Surpassing ideal time-frequency magnitude masking for speech separation,” IEEE ACM Trans. Audio Speech Lang. Process., vol. 27, no. 8, pp. 1256–1266, 2019.
>
> [4] Alexandre D ́efossez, Gabriel Synnaeve, and Yossi Adi, “Real Time Speech Enhancement in the Waveform Domain,” in Proc. Interspeech, 2020, pp. 3291–3295.
>
> [5] Jun Chen et al., “Inter-subnet: Speech enhancement with subband interaction,” in Proc. ICASSP. IEEE, 2023, pp. 1–5.
>
> [6] Simon Welker, Julius Richter, and Timo Gerkmann, “Speech Enhancement with Score-Based Generative Models in the Complex STFT Domain,” in Proc. Interspeech, 2022, pp. 2928–2932.

---

> > ### Comment · Reviewer_KgQK · 2023-11-22
> >
> > Thanks for the response, and I have no further questions. While I will raise the individual score, I will leave my score as it is.

---

### Official Review · Reviewer_VMQy · 2023-11-04

**Soundness:** 2 fair
**Presentation:** 1 poor
**Contribution:** 3 good
**Rating:** 5
**Confidence:** 4

**Summary:**

This paper proposed a new way to encoding speech, by using continuous feature to capture acoustic and discretized token to capture semantics. Based on this mixed codec, a new framework been proposed to to speech generation task. Downstream task include TTS, voice conversion and speech to speech translation.

**Strengths:**

(1) I would like to highlight one concept the paper proposed which is different than previous audio lm based work, it's not necessarily to discretize both acoustic and semantic information. A combined approach (discretize and continuous) might also work.

(2) I covers many different applications, and the framework is easy to adopt.

**Weaknesses:**

(1) The paper is poorly written. Section 3.2 is very hard to understand. I would suggest anything in the figure, should clearly defined in the method section. For example, "codec decoder". I also highly suggest the author describe tortose (Betker, 2023) in the related work. It seems some component borrowed from here, but it's unclear which part is being used.

(2) The way to disentangle acoustic and semantic are very empirical, e.g. assume 6-layer of wav-lm, assume mean capture speaker information. There is no solid evidence to justify those claim.

(3) Results are unsound. It keep mention low bit rate, but I even don't know what the bit rate used here. From the demo, the reconstruction audio quality are poor, which make it hard to believe the proposed codec are really working.

**Questions:**

"First, these codec tokens usually contain as many speech attributes as
possible to high-fidelity reconstruction quality, which leads to information redundancy and increases
the difficulty of predicting the tokens in downstream tasks...."

One key contribution of the paper is replace acoustic tokens in audioLM. But those claim are very vague, are there any empirical or theoretical analysis to justify this claim?

"a neural vocoder is used to reconstruct speech waveforms based on the extracted speech vectors vec
since the vocoder can produce waveforms that are nearly indistinguishable from recorded waveforms
and are highly generalizable outside of the training set (Betker, 2023). "

I highly recommend the author fix citation like this, what are citing here? the claim or the decoder architecture or something else?

---

> ### Author Response · Authors · 2023-11-22
> **Author Response to Reviewer VMQy (1/2)**
>
> Q: The paper is poorly written. Section 3.2 is very hard to understand. I would suggest anything in the figure, should clearly defined in the method section. For example, "codec decoder". I also highly suggest the author describe tortoise (Betker, 2023) in the related work. It seems some component borrowed from here, but it's unclear which part is being used.
>
> A: We have revised section 3.2 and described TorToise-TTS (Betker, 2023) in the related work. We borrowed the contrastive Language-Voice pre-trained (CLVP) module from Tortoise-TTS. Thank you for your suggestion. We have carefully polished the writings and the figures in the updated manuscript.
>
> Q: The way to disentangle acoustic and semantic are very empirical. assume 6-layer of wav-lm, assume mean capture speaker information.
>
> A: As demonstrated by KNN-VC [1], the 6-layer of WavLM is enough to reconstruct high-quality waveforms. Besides, KNN-VC proves that the nearby 6-layer WavLM features have similar phonetic content, which is helpful in obtaining semantic tokens. Figure 2 in Appendix A.2 demonstrates the strong correlation between the averaged vectors and the speaker identities. In addition, the way to disentangle acoustic and semantic is achieved by the combination of mean normalization and K-means algorithm. "mean capture speaker information" can be found in the work [2].
>
> Q: Results are unsound. It keep mention low bit rate, but I even don't know what the bit rate used here.
>
> A: We keep mentioning a low bit rate because previous speech tokens are complex and have a high bit rate. The tokens in NLP have a lower bit rate compared to speech tokens, and LMs in NLP have boosted. Therefore, we try to simplify speech tokens. On the one hand, we decouple the non-linguistic information from speech tokens, since the non-linguistic information is linguistic-irrelevant and not suitable for language modeling. On the other hand, we reduce the bit rate and approach that of NLP tokens. Thus stimulate the potential of speech LM through these methods. Obtaining fewer speech tokens attracts attention [5,6].
>
> Q: One key contribution of the paper is replace acoustic tokens in audioLM. But those claim are very vague, are there any empirical or theoretical analysis to justify this claim?
>
> A: There are many articles that claim the acoustic tokens are too complex to predict for Speech LMs [3,4,5]. Therefore, Mel or continuous vectors are utilized to replace acoustic tokens in current works[3,7]. In addition, kNN-VC[1] demonstrated that speech waveforms can be reconstructed from features of WavLM. We follow these works using WavLM features to replace acoustic tokens.
>
> Q:  I highly recommend the author fix citation like this, what are citing here? the claim or the decoder architecture or something else?
>
> A: The citation here intends to confirm that the waveforms generated by vocoders are highly generalizable outside of the training set. Thank you for your suggestion. We have carefully polished the writings and the figures in the revised manuscript to avoid misunderstandings. We have updated our manuscript and fixed these citations.
>
> [1] Baas, M., Van Niekerk, B., & Kamper, H. (2023). Voice Conversion With Just Nearest Neighbors. ArXiv. /abs/2305.18975
>
> [2] Krishna, V., Sai, T., & Ganapathy, S. (2023). Representation Learning With Hidden Unit Clustering For Low Resource Speech Applications.ArXiv. /abs/2307.07325
>
> [3] Shen, K., Ju, Z., Tan, X., Liu, Y., Leng, Y., He, L., Qin, T., Zhao, S., & Bian, J. (2023). NaturalSpeech 2: Latent Diffusion Models are Natural and Zero-Shot Speech and Singing Synthesizers.ArXiv. /abs/2304.09116
>
> [4] Du, C., Guo, Y., Shen, F., Liu, Z., Liang, Z., Chen, X., Wang, S., Zhang, H., & Yu, K. (2023). UniCATS: A Unified Context-Aware Text-to-Speech Framework with Contextual VQ-Diffusion and Vocoding.ArXiv. /abs/2306.07547
>
> [5] Ren, Y., Wang, T., Yi, J., Xu, L., Tao, J., Zhang, C., & Zhou, J. (2023). Fewer-token Neural Speech Codec with Time-invariant Codes.ArXiv. /abs/2310.00014
>
> [6] Yang, D., Liu, S., Huang, R., Tian, J., Weng, C., & Zou, Y. (2023). HiFi-Codec: Group-residual Vector quantization for High Fidelity Audio Codec.ArXiv. /abs/2305.02765
>
> [7] Nachmani, E., Levkovitch, A., Hirsch, R., Salazar, J., Asawaroengchai, C., Mariooryad, S., Rivlin, E., & Ramanovich, M. T. (2023). Spoken Question Answering and Speech Continuation Using Spectrogram-Powered LLM.ArXiv. /abs/2305.15255

---

> ### Author Response · Authors · 2023-11-22
> **Author Response to Reviewer VMQy (2/2)**
>
> Q: From the demo, the reconstruction audio quality are poor, which make it hard to believe the proposed codec are really working.
>
> A: We are sorry for insufficient training in vocoder, which leads to low reconstruction quality. We have supplemented the speech reconstruction demo and quantitative results of reconstruction quality, which proves our codec is effective. We test the PESQ and STOI metrics in 100 speech clips from our test set. The quantitative results are shown as follows.
>
> | Model | bitrate | PESQ | STOI |
> | --- | --- | --- | --- |
> | Encodec | 1.5kbps | 1.42 | 0.84 |
> | Encodec | 3kbps | 1.87 | 0.90 |
> | WavLM feature +Vocoder | - | 2.50 | 0.95 |
> | WavLM feature +KMeans300 +Inverse k-means +Vocoder | ~0.41kbps | 1.93 | 0.92 |
>
>
> The results indicate that our codec is capable of reconstructing speech with a low bitrate and minor loss of sound quality.

---

> > ### Comment · Reviewer_VMQy · 2023-11-22
> > **Thank you for the detailed reply.**
> >
> > Some of my concern has been addressed. For bitrate, I'd suggest compare with AudioLM or Hubert.
> >
> > I'd like to update my score, but give large part of paper need changed, I still feel it's not good enough for ICLR, and better to revise and submit to next conference.

---

### Official Review · Reviewer_G28A · 2023-11-08

**Soundness:** 3 good
**Presentation:** 4 excellent
**Contribution:** 3 good
**Rating:** 8
**Confidence:** 4

**Summary:**

The paper proposes an encoder-decoder architecture that decouples acoustic style and semantic linguistics. The semantic tokens can be combined with prompts for LLMs to generate output token sequence which could be further combined with optionally modified acoustic style vectors to reconstruct speech for various speech tasks. The authors conducted experiments in VC, TTS, and S2ST to demonstrate that their proposed system can match or outperform some recent models on those tasks.

**Strengths:**

Originality: Decoupling or disentangling speech style and speech content with neural networks has been long studied since VAE and GAN. The authors are able to leverage recent developed models such as WavLM to extract speech vectors for downstream speech generation tasks. Using K-mean to cluster speaker vectors and using BPE to compress token sequences are also popular techniques in related publications, but the authors also incorporate them as essential components in their speech generation system. They also proposed inverse K-means that includes speaker vectors as part of the prompts to generate speech vectors rich in speaker style.

Quality: Despite the novel engineering integration and encouraging experimental results, the authors did not attempt to develop their proposed system with a more theoretical approach. As a result, readers may not be able to gain as many insights as to why the proposed system and its components can outperform the competing systems.

Clarity: The paper is well-written and easy to follow. There are few errors. The authors used high level equations to describe their system design. The use of diagrams and coloring schemes are appropriate. The result tables contain the right amount of information for the readers. The demos on the github sites are well-organized.

Significance: The paper engages the latest trend of speech research in the community by bridging the use of LLM into speech generation.

**Weaknesses:**

A few sections are written without adequate explanation of the symbols or extensive reader knowledge is assumed beyond a reasonable context. For example, in Equation 6, a more detailed description of the adversarial setup should be given. Why is the loss constructed this way? Why is MPD and MSD selected? The equation of the feature matching loss (not named as such in the reference) and reconstruction loss (which norm?) should also be clearly stated.
Minor typo below equation 9. ^vec instead of ^wave. Less well-known acronyms such as CLVP in section 4 should be expanded first. The experimental section should also compare the model sizes of the proposed system and the competing baselines.

**Questions:**

In Figure 1, what's the use of two codec encoders for the TTS pipeline?
Did the author compare the performance of TTS and S2ST with AudioPaLM?

---

> ### Author Response · Authors · 2023-11-22
> **Author Response to Reviewer G28A**
>
> Q: Why is the loss constructed this way? Why is MPD and MSD selected? The equation of the feature matching loss (not named as such in the reference) and reconstruction loss (which norm?) should also be clearly stated.   Minor typo below equation 9. ^vec instead of ^wave. Less well-known acronyms such as CLVP in section 4 should be expanded first. The experimental section should also compare the model sizes of the proposed system and the competing baselines.
>
> A: The loss, MPD, and MSD follow the setting of HiFiGAN [1], as it is a classic and powerful vocoder.  Thank you for constructive comments, and suggestions. We have carefully polished the writings and the figures in the updated manuscript.
>
> Q: In Figure 1, what's the use of two codec encoders for the TTS pipeline?  Did the author compare the performance of TTS and S2ST with AudioPaLM?
>
> A: One codec encoder of the TTS pipeline is used to extract semantic tokens that are used as prompts in LM to control the speaking style of synthetic speech, and the other is used to extract speech vectors that are used as prompts in the inverse K-means module to control the speaker timbre of synthetic speech. As there is no open-source code or released model of AudioPaLM, we did not compare the performance of TTS and S2ST with AudioPaLM.
>
> [1] Kong, J., Kim, J., & Bae, J. (2020). HiFi-GAN: Generative Adversarial Networks for Efficient and High Fidelity Speech Synthesis.ArXiv. /abs/2010.05646

---

### Official Review · Reviewer_cYW2 · 2023-11-10

**Soundness:** 3 good
**Presentation:** 2 fair
**Contribution:** 2 fair
**Rating:** 5
**Confidence:** 4

**Summary:**

In this paper, the authors propose a new generative speech network that combined conditioning on discrete tokens for semantic content with conditioning on continuous features for audio style. They use a pre-trained WavLM model as a source of both features where the discrete features are obtained via K-means clustering WavLM features, and the continuous features are encoded directly from WavLM features. During training, a language model is trained on BPE encoded discrete token sequences, while a decoder is trained on a combination of discrete and continuous inputs where the former are processed via an "inverse K-means" process. The resulting output is fed into a vocoder to produce speech.

The authors show that such a model can perform several tasks very well, including voice conversion, zero shot speaker style transfer TTS and speech to speech translation.

**Strengths:**

- Strong results on several benchmarks across several tasks
- Novel architecture that utilizes discrete and continuous features and avoids multi-pass decoding like VALL-E/SoundStorm

**Weaknesses:**

- Quite complex and depends on previous pre-trained models (e.g. WavLM)
- No comparison with diffusion based models like NaturalSpeech2
- Missing ablations make it difficult to understand which parts contribute to the performance of the model. While certain parts of the architecture are ablated for certain tasks, it would be nice to ablation results for each task to understand how modeling choices (BPE, inverse K-means) contribute to the performance. It would also be good to know how much WavLM contributes to the overall performance or if it is replaceable with any other model. Choices on the LM side (how necessary it is to produce 256 candidates and then rank them? etc) are not ablated at all.
- Speech to speech translation results compare to one other prior work that outperforms the proposed architecture on BLEU and is missing all other metrics making comparison difficult.

**Questions:**

Most of my questions relate to the weaknesses section:
- Could you provide additional ablations that can make it clear what modeling choices lead to what outcomes?
- Could you provide a more thorough comparison on the STST task?
- Could you ablate the choice of relying on WavLM as the primary encoder in this method?

---

> ### Author Response · Authors · 2023-11-22
> **Author Response to Reviewer cYW2 (1/2)**
>
> Q: No comparison with diffusion based models like NaturalSpeech2
>
> A: Indeed, it makes more sense to compare the diffusion model. However, since there is no open source code, we do not get a reasonable performance when we reproduce diffusion models  during experiments such as naturalspeech2 or voicebox. Therefore, we did not compare diffusion models.
>
> Q: Could you provide additional ablations that can make it clear what modeling choices lead to what outcomes?
>
> A: Thank you for your suggestion. Since we propose a multi-task framework, we validated the proposed module in several specific tasks, such as verifying the validity of inverse k-means in zero-shot VC and BPE in zero-shot TTS in Table 1, 2, and 3.  We supplement the ablation of the CLVP module and the WavLM model. The results are shown in next comment.
>
> In the process of our ablation experiments, we found that CLVP can reduce the frequency of extremely poor samples, which in turn increases the stability of the generation, and after removing the CLVP, we found that the extremely poor samples will affect the evaluation indexes to a large extent, so we reduced the number of samples selected by CLVP for ablation. As for the pre-training representation selection, we found that different representation selections have little effect on the model performance.
>
> Q: Speech to speech translation results compare to one other prior work that outperforms the proposed architecture on BLEU and is missing all other metrics making comparison difficult. Could you provide a more thorough comparison on the S2ST task?
>
> A: We try to compare our model with other S2ST models. However, the models either involve internal datasets [1,2] or have complex pre-training and fine-tuning pipelines [3,4]. The model trained on the same open-source  dataset as ours is GigaST [5]. However, it is a speech-to-text translation (S2TT) model; therefore only BLUE is available and comparable. As our model is a text-free S2ST model and transfers source speaking style and speaker timbre when translating, it is reasonable to obtain a lower BLEU score than an S2TT model (GigaST).
>
> Q: Quite complex and depends on previous pre-trained models (e.g. WavLM). Could you ablate the choice of relying on WavLM as the primary encoder in this method?
>
> A: Yes, we ablate WavLM during the review period. We choose the 6-layer feature of the XLSR model, which is a wav2vec 2.0 model pre-trained on a multilingual dataset. We find the performance of XLSR is similar to that of WavLM. We speculate that the shallow-layer feature has high generalization ability for various speech inputs [6].
>
> [1] Song, K., Ren, Y., Lei, Y., Wang, C., Wei, K., Xie, L., Yin, X., & Ma, Z. (2023). StyleS2ST: Zero-shot Style Transfer for Direct Speech-to-speech Translation.ArXiv. /abs/2305.17732
>
> [2] S. Popuri, P. Chen, C. Wang, J. Pino, Y. Adi, J. Gu, W. Hsu, and A. Lee, “Enhanced direct speech-to-speech translation using selfsupervised pre-training and data augmentation,” in Interspeech 2022. ISCA, 2022, pp. 5195–5199.
>
> [3] Communication, S., Barrault, L., Chung, Y., Meglioli, M. C., Dale, D., Dong, N., Duquenne, P., Elsahar, H., Gong, H., Heffernan, K., Hoffman, J., Klaiber, C., Li, P., Licht, D., Maillard, J., Rakotoarison, A., Sadagopan, K. R., Wenzek, G., Ye, E., . . .  Wang, S. (2023). SeamlessM4T: Massively Multilingual & Multimodal Machine Translation.ArXiv. /abs/2308.11596
>
> [4] S. Nakamura, K. Markov, H. Nakaiwa, G. Kikui, H. Kawai, T. Jitsuhiro, J. Zhang, H. Yamamoto, E. Sumita, and S. Yamamoto, “The ATR multilingual speech-to-speech translation system,” IEEE Trans. Speech Audio Process., vol. 14, no. 2, pp. 365–376, 2006.
>
> [5] Ye, R., Zhao, C., Ko, T., Meng, C., Wang, T., Wang, M., & Cao, J. (2022). GigaST: A 10,000-hour Pseudo Speech Translation Corpus.ArXiv. /abs/2204.03939
>
> [6] Yang, S., Chi, P., Chuang, Y., Lai, C., Lakhotia, K., Lin, Y. Y., Liu, A. T., Shi, J., Chang, X., Lin, G., Huang, T., Tseng, W., Lee, K., Liu, D., Huang, Z., Dong, S., Li, S., Watanabe, S., Mohamed, A., . . .  Lee, H. (2021). SUPERB: Speech processing Universal PERformance Benchmark.ArXiv. /abs/2105.01051

---

> ### Author Response · Authors · 2023-11-22
> **Author Response to Reviewer cYW2 (2/2)**
>
> Table 1: Results of inter-lingual zero-shot VC with 95% confidence interval.
> | Model | Naturalness MOS | Speaker Similarity MOS | CER(%) | WER(%) | SCS |
> | --- | --- | --- | --- | --- | --- |
> | GroundTruth | 4.42±0.07 | - | 2.8 | 1.9 | - |
> | LM-VC | 3.78±0.10 | 3.74±0.08 | 3.2 | 2.7 | 0.892 |
> | Vec-Tok Speech | 3.86±0.10 | 3.93±0.09 | 3.0 | 2.6 | 0.927 |
> | w/o Inv_K | 3.88±0.08 | 3.67±0.08 | 2.9 | 2.4 | 0.878 |
> | w/ XLSR | 3.85±0.11 | 3.89±0.07 | 3.0 | 2.7 | 0.928 |
>
> Table 2: Results of cross-lingual zero-shot VC with 95% confidence interval.
> | Model  | Naturalness MOS | Speaker Similarity MOS | CER(%) | WER(%) | SCS |
> | --- | --- | --- | --- | --- | --- |
> | GroundTruth | 4.42±0.07 | - | 2.8 | 1.9 | - |
> | LM-VC | 3.60±0.11 | 3.66±0.08 | 3.9 | 3.1 | 0.892 |
> | Vec-Tok Speech | 3.81±0.10 | 3.90±0.09 | 3.2 | 2.9 | 0.919 |
> | w/o Inv_K | 3.79±0.09 | 3.63±0.10 | 3.4 | 2.8 | 0.871 |
> | w/ XLSR | 3.80±0.08 | 3.87±0.09 | 3.1 | 3.0 | 0.922 |
>
> Table 3: Results of zero-shot TTS with 95% confidence interval.
> | Model  | Naturalness MOS | Speaker Similarity MOS | Style Similarity MOS | CER(%) | WER(%) | SCS |
> | --- | --- | --- | --- | --- | --- | --- |
> | GroundTruth | 4.40±0.07 | - | - | 2.8 | 1.9 | - |
> | VALL-E X | 3.72±0.12 | 3.67±0.10 | - | 5.3 | 3.9 | 0.852 |
> | Bark | 3.68±0.11 | 3.73±0.12 | - | 5.8 | 3.6 | 0.866 |
> | Vec-Tok Speech | 3.92±0.08 | 3.87±0.07 | - | 3.7 | 3.4 | 0.909 |
> | w/ Style Prompt | 3.88±0.10 | 3.64±0.08 | 3.87±0.10 | 3.8 | 3.7 | 0.902 |
> | w/o BPE | 3.80±0.10 | 3.88±0.08 | - | 4.6 | 4.2 | 0.908 |
> | w/ CLVP-64 | 3.84±0.12 | 3.86±0.07 | - | 4.4 | 4.1 | 0.907 |

---

### Author Response · Authors · 2023-11-22
**General Response**

We sincerely thank the efforts of all reviewers, for their valuable, professional, and constructive comments.

We updated our manuscript to address the issues raised by reviewers. We have made the following changes to the paper:
- We have carefully polished our citations and writings in our revised manuscript.
- We discuss Tortoise-TTS in related work and provide a detailed description of Vec-Tok Codec in section 3.2.
- We have supplemented the additional ablations of CLVP and the pre-trained SSL model. The results and analysis are shown in sections 5.1 and 5.2.
- We have updated the quantitative results of speech denoising and speech reconstruction in Appendix A.3 and Appendix A.4.

We also updated our demo page to demonstrate the reconstruction quality of our proposed framework. The updated part is available here: https://vectokdemo.github.io/VecTok/#Recon

---

### Meta-Review · Area_Chair_uAfW · 2023-12-02

**Metareview:**

In this work, the authors describe al speech generation model, dubbed d Vec-Tok Speech, where a codec uses continuous feature to capture acoustic and discretized token to capture semantics.  The proposed work  tacks some shortcomings of the previous works related to neural audio codec. Despiste the authors have addressed several major concerns in the rebuttal phase,  there present version of the paper still needs a major revision. The novelty appears to be incremental; furthermore, there are still concerns about the audio quality.

**Justification For Why Not Higher Score:**

The document still needs a major revision. There are elements of originality, but the overall novelty is incremental.  The audio quality seems to still not be good enough.

**Justification For Why Not Lower Score:**

N/A

---

### Decision · Program_Chairs · 2024-01-16

Reject